# Piezoelectric row-column sensing system on table tennis rackets for hit and rotation measurement

Chao Zhang[1], Zile Fan[2], Yafeng Kang[3]*

1 Shanghai University of Finance and Economics Zhejiang College, Jinhua, Zhejiang, China, 2 School of Marxism Studies, Yiwu Industrial and Commercial College, Yiwu, Zhejiang, China, 3 Xingzhi College, Zhejiang Normal University, Jinhua, Zhejiang, China

* full121314@163.com

## Abstract

Wearable sensing systems are often constrained by the number of available signal channels: expanding the sensor number typically improves human motion monitoring but drives up hardware complexity. In this paper, a row-column sensing method is proposed to address this limitation in the context of table tennis impact monitoring. The hit sensing position on the racket is decomposed into orthogonal row and column coordinates, with only one single striped piezoelectric PVDF (Polyvinylidene fluoride) flexible sensor placed on each axis. By combining the signals on each row and column, the hitting position is analytically obtained. In a 5 × 5 layout this architecture reduces required signal pathways from 25 to 10 (five rows plus five columns) while delivering same spatial accuracy. Additionally, the denser set of impact locations also enables detection of ball spin. In-plane and out-of-plane rotations produce distinct stress distributions across the racket surface, which the array captures through differential row and column signal patterns. This approach can be extended to other wearable or sports devices that need higher spatial resolution without proportional increases in channel count, and it shows clear potential for advancing table tennis training, officiating, and performance analysis.

## Introduction

Table tennis is one of the most widely enjoyed sports in the world. No matter in terms of competitive sports or physical exercise, it is wildly welcomed and has broad global audiences [1,2]. As a result, the exploration of table tennis techniques has always been a topic of interest for researchers [3]. Although vision-based method can detect and forecast the table tennis movement trajectory effectively, the contact between table tennis racket and the ball, such as hitting location, hitting force, and ball spin, contain many useful information, which is important to the table tennis training, referee, and development [4]. Recent advances in flexible piezoelectric materials

**Data availability statement:** All relevant data are within the paper and its Supporting Information files.

**Funding:** The author(s) received no specific funding for this work.

**Competing interests:** The authors declare that there is no conflict of interest.

provide a means to monitor these traits because of the development in innovative materials [5,6] and excellent wearability and mechanical detection sensitivity [7,8]. In sports monitoring, flexible piezoelectric materials have been applied to areas such as injury prevention [9], badminton monitoring [10], running gesture analysis [11], and basketball training [12,13] etc.

While piezoelectric materials have been utilized in table tennis detection, there are two key challenges that hinder the precision of detection and broader application. First, there is a limitation in the number of detectable points [14,15]. The area of a table tennis racket is approximately 240 square centimeters and the size of ball is about 38 mm [16]. If only 10 detection points are set, each point would cover roughly 24 cm² , which is quite large compared to the size of a sensor, making precise measurements difficult [17,18]. Adding more sensing points improves the sport analysis such as stroke evaluation and tactical assessment [19,20]. However, current detection technologies typically associate each detection point with a transmission channel, meaning that increasing the number of detection points requires scaling up the hardware, which greatly complicates the monitoring system and integration [21,22]. The second challenge is the spin measurement. The skill level in table tennis depends not only on the force and placement of the shot but also on the spin of the ball [23]. A ball with a higher spin speed is more challenging to defend against [24]. Therefore, measuring spin is crucial, yet direct spin sensing remains underexplored in the existing systems.

This study addresses the two limitations identified above with the solution inspired by the analysis of the ball-reception process in table tennis. Traditional flexible detection systems typically place one discrete sensor in each cell of a single-layer grid. Expanding the sensor number therefore requires proportional increases in channels, circuitry, and communication modules, rapidly escalating system complexity [25,26]. The purposed method decomposes the table tennis racket face into orthogonal rows and columns and mount one long, strip-shaped sensor along each. The impact coordinates are obtained by combining the voltage responses of the active row and column. This row–column strategy reduces sensor (and channel) count from $m \times n$ to $m + n$ while expanding the number of resolvable points and lowering hardware cost. Spin detection is integrated within the same framework. Although table tennis employs varied spins (topspin, backspin, sidespin, combinations), they can be grouped by whether rotation lies primarily in the plane of the racket or out of that plane [27,28]. The interaction between a spinning ball and a racket generates produces asymmetric, spin-dependent local deformation and stress around the contact zone on the rubber layer. With sufficient spatial distribution of sensors, these differential stress patterns allow simultaneous estimation of impact location, force magnitude, and spin characteristics [29,30]. We implement this row–column flexible piezoelectric array on a racket, achieve accurate point localization with markedly fewer channels, and present, to our knowledge, the first preliminary direct measurements of table tennis spin using an embedded flexible sensing network. The following sections detail the sensor design, signal processing, and experimental validation.

The paper is arranged according to this structure: we firstly address the background and difficulties of table tennis monitoring and introduces the proposed method and principles of the piezoelectric row-column electrodes on table tennis racket. After that, the details of the design are explained and we measured the design by hitting of table tennis ball with and without rotation. Between them, we also do the simulation investigation on the influence of the rotation to the racket. At last, we conclude the work and provide the discussion of the future development. This device can be used in monitoring of table tennis game, guide for the sports training, and future design of sport game robot.

## Design and methods

This paper has two main objectives: (1) to exploit row–column sensor interactions within a fixed racket area to create a finer virtual division of detection points, thereby enabling more accurate measurement of forces applied to the racket surface; and (2) to demonstrate a preliminary capability for measuring table tennis ball spin. As illustrated in Fig 1, when a player hits the ball, the racket imparts force that alters the ball's motion, while the ball simultaneously exerts a reaction force on the racket. If the time-resolved forces originating at different surface positions can be acquired, the racket–ball contact event can be characterized; during this event, the effective contact region is on the order of ~$10^{-2}$ cm² and the duration is on the millisecond scale [31]. A denser spatial distribution of sensing points therefore enables more precise localization and force reconstruction [32]. Furthermore, in-plane and out-of-plane spins induce distinct stress propagation patterns on the racket, even if the differences are subtle. As shown schematically on the left of Fig 1, out-of-plane spin (rotation axis mainly in the racket plane) tends to produce a predominantly linear propagation of force along a row or column, activating multiple collinear strip sensors near the impact point [33]. In contrast (right of Fig. 1), in-plane spin

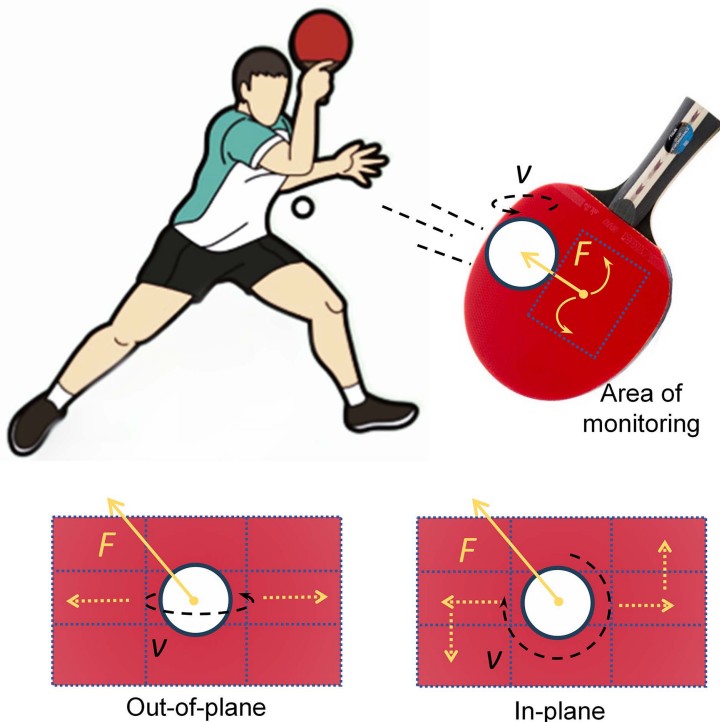

**Fig 1. Demonstration of the hitting process of table tennis on racket.** The bottom line shows the interaction of racket and ball with different rotation. The out-of-plane rotation generated force is worked on sensors on a line getting across the hitting position, while the in-plane rotation causes the generated signal not only on a row or in a column.

(rotation axis roughly normal to the racket) produces a more two-dimensionally dispersed distribution, so responsive sensing elements are not confined to a single row or column. With a higher-resolution measurement matrix, these spin-dependent patterns become, in principle, distinguishable.

Fig 2 illustrates the conventional point-to-point sensor layout and the proposed row–column sensing array. In the traditional point-scanning design, each sensor requires its own dedicated signal line; thus, an array with $n$ rows and $m$ columns needs $m \times n$ signal channels. If the hitting position is $(i, j)$, only the sensor at that position will capture the signal. In the proposed row–column architecture, orthogonal strip sensors share lines: an impact at $(i, j)$ simultaneously activates the i-th row strip and the j-th column strip, so the impact location is determined by intersecting the active row and column. This reduces the required number of channels to $m+n$ while retaining $m \times n$ sensing points. In our real in-racket test, $m=n=5$, yielding 25 sensing points with only 10 channels. The lower part of Fig 2 shows the cross-section: two orthogonal layers of five strip-shaped PVDF piezoelectric sensors each (LOXGO B0D4LG49LJ) [34,35]. To eliminate the crosstalk noise between the transducers, a thin hard bind layer is put between sensors. On the top of the system, a protection layer is fixed; the racket containing the rubber surface and wood substrate is put below the system. Each strip-shaped sensor has a signal output channel, which will transmit the signal to the signal processing device. Here, the MCU (ESP32 microprocessor) powered by a button coin cell battery (LR44 Alkaline) is used to collect signals from the sensor with sampling rate of 24 MHz for signals from all the sensors. Then the signal is transmitted to the personal computer by a low-power, long-range communication module (ATK-LORA-01, 410–441 MHz, Zheng Dian Co. Ltd).

Fig 3 shows the fabricated on-racket impact sensing system. From top to bottom, the system consists of the following layers: a vinyl film protection layer, the top PVDF row sensor layer (with 1 cm spacing between each sensor), a hard bind made of a 0.2 mm thick iron net, the bottom PVDF column sensor layer, and the racket itself. The hard bind is crucial for

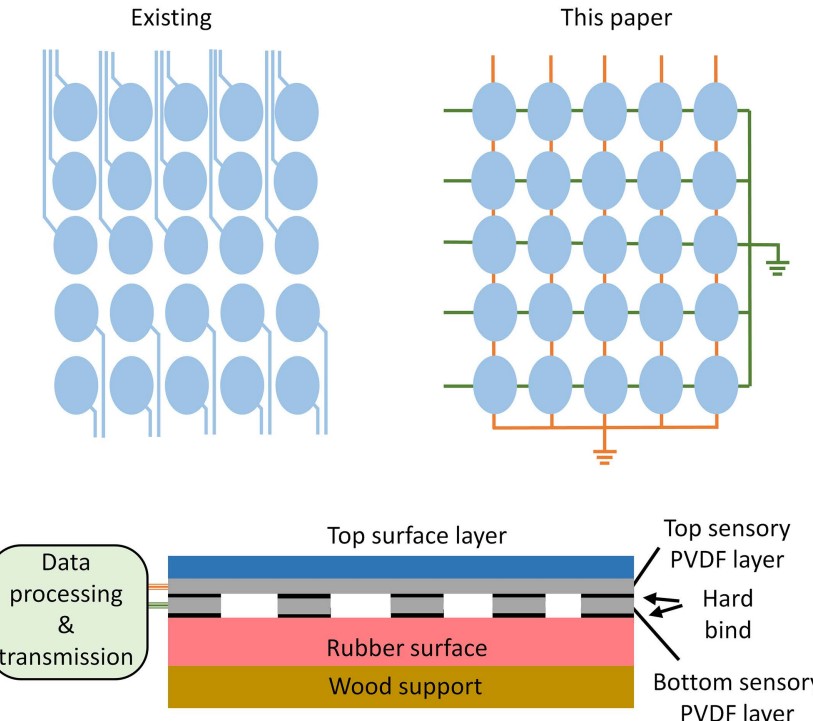

**Fig 2. Demonstration of the existing point array design and proposed row-column piezoelectric sensing array design.** The bottom figure is cross-section demonstration of the layered sensing structures on table tennis racket.

eliminating crosstalk effects [36,37]; without it, the layers would be in direct contact, and force applied to one hitting position could easily induce signals in other positions. By incorporating a square-shaped iron net (approximately 1 cm × 1 cm in size), the hitting force applied to the top layer only induces a signal in the corresponding bottom layer position, provided no rotational forces are introduced to the system.

However, even with this structure the signals from the top and bottom sensor layers differ for the same impact. To quantify and compensate for this difference, we performed a calibration using a miniature electrodynamic shaker (Model K2002E01) that applied a 10 Hz vibration to the assembled system (left side of Fig 4). Both layers produced clear piezoelectric responses, but their amplitudes differed because of the unequal transmission path thicknesses. Representative waveforms from the two layers are plotted on the right side of Fig 4. The bottom-layer signal amplitude is roughly one-third of the top-layer amplitude, yet it remains well above the detection threshold for impact monitoring. These absolute amplitudes are subsequently used for normalization to correct for the inter-layer magnitude disparity. Although minor timing offsets and slight waveform shape differences appear—attributable to mechanical and structural variations—the principal signal features (overall shape and dominant frequency content) are consistent, confirming effective system operation [38].

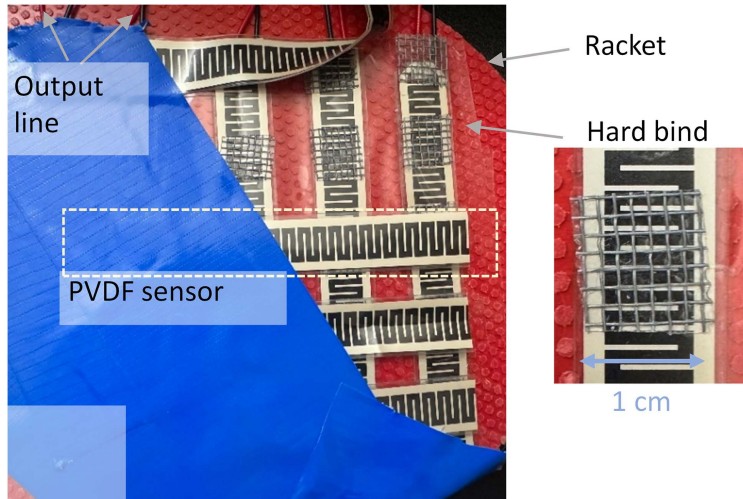

**Fig 3. The picture of the fabricated on-racket hit sensing system.**

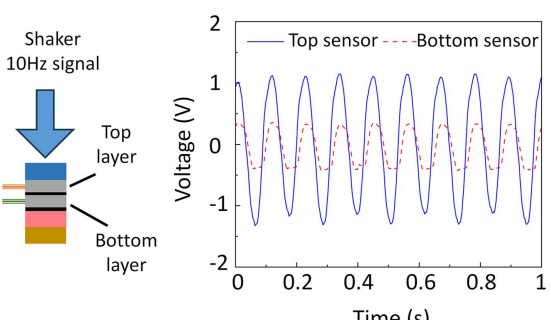

**Fig 4. Demonstration of the force-generated signal calibration test and the captured signal from top and bottom layer.**

## Results

A real table tennis hitting experiment was conducted using the fabricated measurement system, and the resulting signals were recorded. The measured signals from four hits, along with an illustration of the 5×5 sensor array, are shown in Fig 5. The sensor array consists of only 10 striped sensors (compared to 25 for 5×5 sensor array in the previous design method). During each hit, signals are generated from both the top and bottom sensor layers, allowing the hitting position to be determined at the inter-section of the maximum voltage signals from both layers. For instance, in the first hit shown in Fig 5, the maximum voltage from the top layer occurs at sensor T2, while the maximum voltage from the bottom layer occurs at sensor B2. The hitting position is then identified as the sensor at the intersection of column two and row two, as illustrated in Fig 5. The other hit positions can also be traced by this method, demonstrating the efficiency of it. It is also

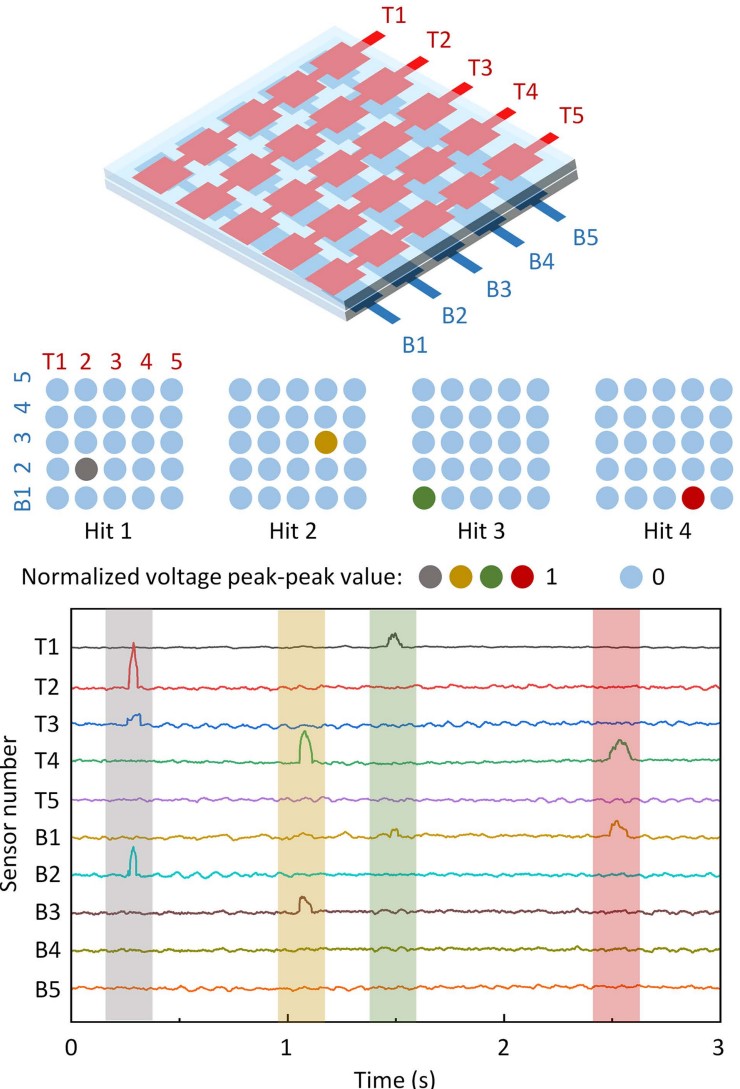

**Fig 5. Illustration of the structure of the sensor array applied in the real monitoring (left) and its ability for multipixel detection.** The array contains 5×5 sensors, which is able to monitor 25 sensing positions. The pressure stimuli (top right) and the corresponding real-time signal (bottom right, unit: V) from the sensor array show four hits from table tennis at different time, marked with different colors.

reasonable to find that the signal from bottom layer will always generate a smaller signal because the hitting pressure will be reduced during the transmission. After the processing using normalization parameters obtained in Fig 4, the different signal will reach the similar level. Through the real-game testing, we did not observe any apparent hitting caused table tennis motion change because of the existence of hard bind layer. To minimized the potential influence for monitoring system on gameplay, several methods can be implemented in the commercialized monitoring system, for example, design a structured sunken rubber layer to accommodate the hard bind to make the surface flat, increasing the thickness of top layer, or reducing the size of monitoring layer.

Another important task for this design is to investigate the possibility of rotation measurement. A simulation using different table tennis inputs is first performed to investigate the stress distribution with different rotated table tennis hits, where the setup and results are shown in Fig 6. In table tennis sport, there are many different surface layer designs. For example, the inverted rubber has a more plat and soft surface to make the ball have better rotation, while pimpled rubber can provide stronger hitting force to make the ball have quick movement. In this paper, we simplify the design and only use a layer of rubber material to investigate. The simulation is conducted in COMSOL 6.0 software using the mechanics module. A layer of 5 mm rubber is used to represent the surface of the racket. A same magnitude 5 N force is added in the center of tubber sheet. To simulate the rotation caused by the table tennis ball hitting, the force is set as different input directions for out-of-plane rotation in the left and in-plane rotation in the right side. The out-of-plane rotation only give a force along the $y$-axis inside $xy$-plane, where its magnitude depends on the input angle. Here the angle is 30° and the $F_{out\_y}$ is 2.5 N. The in-plane rotation enforces a rotated torque in $xy$-plane. As a result, the $F_{in\_xy}$ is set as 1.78 N at $x$ direction and 1.78 N at $y$ direction. These magnitudes are calculated by a 45° azimuth angles, which can also be adjusted to represent different ration manner. Because the property of used piezoelectric material, only z-direction force will generate an electric voltage signal, where the normalized force distributions are plotted in Fig 6. The in-plane force result has a single-direction force intensity distribution. For the in-plane rotation, the force distribution is more unsymmetric in the $xy$-plane: the force will not conduct in the $y$-direction but also transmit along the $x$-direction. As a result, when the table tennis has different

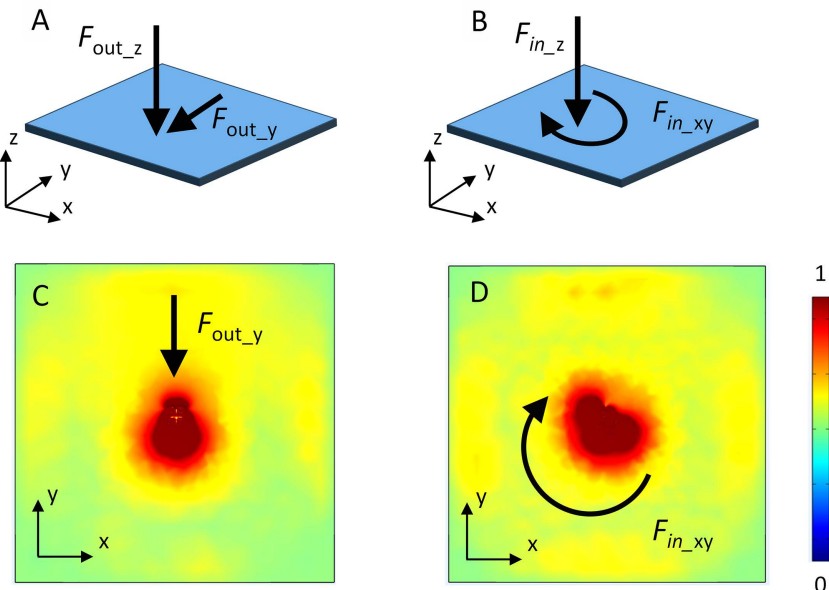

**Fig 6. Simulated z-direction volume stress distribution on an elastic material (rubber) with out-of-plane force and in-plane force as input: A. demonstration of the decomposed out-of-plane input rotation force.** B. demonstration of the decomposed in-plane input rotation force. C. volume stress distribution with out-of-plane force input. D. volume stress distribution with in-plane force input.

rotation forces as input, the stress distribution on elastic sheets: both surface layer on the monitoring system and rubber layer on the table tennis racket, should have similar stress distribution, while the piezoelectric sensors can capture these signals.

The real test results of the system for rotated table tennis sensing are shown in Fig 7. The era of the signal is picked up from a 50-hitting signal in the real game from two players, in which they are asked to play table tennis using the rotating ball as more as possible. For the signals displayed in Fig 7, they are generated by three hits with different rotations: one from out-of-plane rotation and two from in-plane rotation. For each impact we first window the waveform around the contact (based on the global peak and a fixed pre/post margin), then extract the peak-to-peak voltage for every top strip (T1–T5) and bottom strip (B1–B5). These peak amplitudes are normalized by the maximum value within the respective impact to reduce variability due to overall force. As an example, hit 1 produced (Top) T1 0.05, T2 0.08, T3 1.00, T4 0.06, T5 0.07 and (Bottom) B1 0.02, B2 1.00, B3 0.60, B4 0.04, B5 0.03. We form a virtual 5×5 impact map by the outer product $M(i, j)$

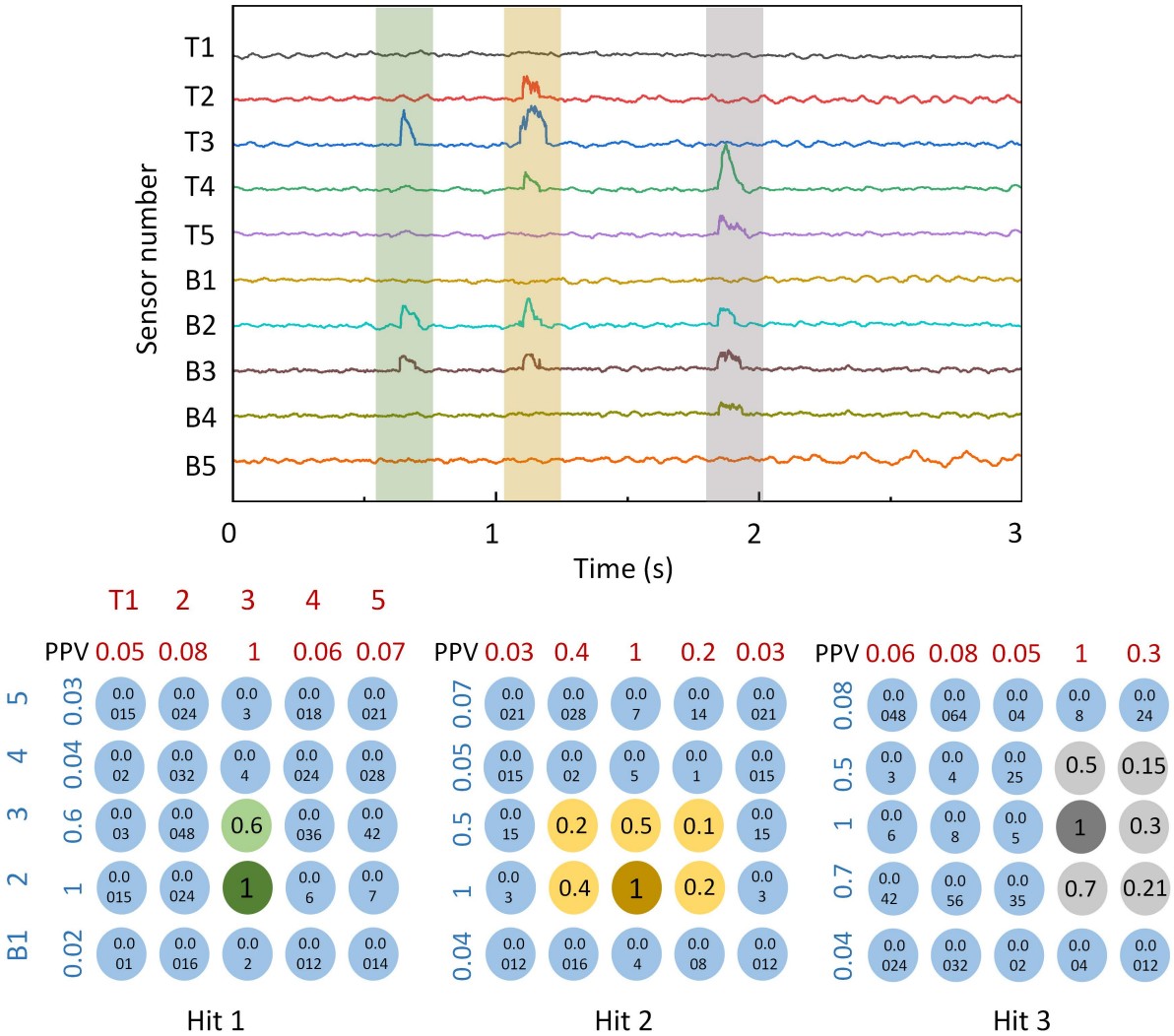

**Fig 7. Experimental time-dependent electronic signal and extracted peak-peak value of the rotated table tennis ball hitting test.** Three hits are recorded, where the first one is an out-of-plane rotation and the other two are in-plane rotation hitting.

= $T\_i \times B\_j$ ($i$ = row from top layer, $j$ = column from bottom layer). For Hit 1 this yields a dominant ridge centered at (row 3, columns 2–3) with minimal spread in the row direction, indicating an out-of-plane rotation of ball with tangential shear primarily along one axis; this matches a single-axis (side-spin–like) case. For the hit 2 after 1 s, energy spreads over T2–T4 and B2–B3 with the peak near (row 3, column 2); the broader multi-row participation together with multi-column activation is consistent with an in-plane spin. Hit 3 near t = 3s shifts the centroid toward higher row index (T4–T5 engaged) and columns B2–B4, again indicating an oblique spin but with the tangential vector rotated, displacing the weighted centroid diagonally. These results show that localization is preserved using only 10 channels for the 10 × 10 cm2 monitored area and spin mode influences the anisotropy and spread of the reconstructed impact map, enabling inference of rotation type from sparse strip data. The approach can be readily scaled to finer granularity by narrowing strip width or adding a second staggered strip set Table 1.

Table I compares representative sports monitoring approaches. We introduce the communication (channel) efficiency, defined here as (number of distinct monitored positions or force-sensing elements effectively resolvable)/ (number of physical communication channels). Most prior systems exhibit low efficiency because each sensing site (or a small cluster) requires a dedicated wired or wireless transmission path. Our row–column sparse addressing attains the highest efficiency among the surveyed works—matched only by Ref. [28], which also uses a 5×5 crossed array to reconstruct force distributions and bending/ stretching—but this study doesn't show a clear application yet. While one might be tempted to equate higher efficiency directly with proportional energy savings (e.g., "2.5× lower power"), such a universal claim is not justified because wireless modules differ markedly in per-channel energy cost. Nevertheless, reducing channel count lowers front-end hardware complexity (fewer amplifiers/ ADC inputs/ MUX lines), simplifies flexible interconnects, and eases miniaturization—advantages that are particularly important for wearable or racket-embedded implementations. Beyond efficiency, most reports focus solely on impact (hit) localization or ball trajectory; these are valuable but provide an incomplete picture of player technique. In Ref. [23], an inertial sensor is mounted on the racket to detect the spin of ball. Although it shows a more than 70% accuracy, it did not provide the hitting position information, where the hitting position should also contribute to the spin detection accuracy. Many other studies concentrate on sensing unit structure [28, 30, 31] or novel transducer materials [39,40] rather than holistic system architecture. To our knowledge, our work is the first having innovation from system level which shows promising prospect to detect more information with improved communication efficiency.

**Table 1. Comparison of the different reported works for sport monitoring, especially table tennis monitoring.**

| Reference | Monitored sport | Monitored movement | Communication channel number (row number + column number) | Monitoring position number | Communication efficiency (monitoring position number/ communication channel number) |
|---|---|---|---|---|---|
| [12] | Basketball | Hitting force position | 9 | 9 | 1 |
| [13] | Basketball | Bending force | 1 | 1 | 1 |
| [14] | Table tennis | Acceleration | 1 | 1 | 1 |
| [23] | Table tennis | Spin video | 1 | 1 | 1 |
| [28] | No real application | Bending & stretching force | 10(5 + 5) | 25 | 2.5 |
| [30] | No real application | In-plane force | 1 | 1 | 1 |
| [31] | Tennis | Ball rolling | 9 | 1 | 0.11 |
| [39] | Table tennis | Hitting angle/velocity | 1 | 1 | 1 |
| [40] | Table tennis | Hitting | 1 | 1 | 1 |
| This work | Table tennis | Hitting position/rotation | 10(5 + 5) | 25 | 2.5 |

## Discussion

This paper proposes a row-column interlaced sensing strategy to replace the past point-to-point measurements, which has been validated for use in table tennis detection. This method significantly enhances signal transmission efficiency, thereby reducing the number and complexity of hardware components in wearable devices. The interference between devices from different rows and columns was not observed, possibly caused by the addition of small gaps between the top and bottom layer sensors and the rigid substrate on the system bottom. This design strategy can be widely applied to various types of wearable devices.

Additionally, with the increased sensor density, we successfully measure the rotation of the table tennis ball. Although the ball's rotation also depends on the player's posture and technique in the real match, measuring the relative rotation between the ball and the racket represents a significant advancement. In this study, we focused on hardware innovation, providing measurements and judgments for hit positions and ball rotation. In the future, artificial intelligence and signal processing can be incorporated to enable more detailed analysis of table tennis movements. For example, the hybrid spin of table tennis with both in-plane and out-of-spin rotation can also be distinguished. Moreover, the current detection system is still relatively bulky, which could interfere with player performance during a match. With future optimization of components and materials, it may be possible to integrate the entire system seamlessly into the racket, make it practical in table tennis training, refereeing, and sports analysis [41].

## Conclusion

This paper proposes a row-column interlaced sensing strategy to replace the past point-to-point measurements, which has been validated for use in table tennis detection. This method significantly enhances signal transmission efficiency, thereby reducing the number and complexity of hardware components in wearable devices. In conclusion, a row-column electrode strategy was applied to develop a system for detecting table tennis hit positions. This method reduces the number of communication signal channels effectively, from $m \times n$ to $m + n$. Additionally, the denser sensor distribution also enables the measurement of table tennis ball rotation. In experiment, the system using row-column electrode strategy is built and verified, which works effectively for hitting position detection, followed by the capture of rotation status from the table tennis ball.

## Supporting information

**S1 Table. Data source of the Fig 4.** The time and measured electrical signal data from top sensor and bottom sensor on a overlapped structure.
(XLSX)

**S2 Table. Data source of the Fig 5.** The time and measured electrical signal data from ten channels of the system with four hits on different positions.
(XLSX)

**S3 Table. Data source of the Fig 7.** The measured electrical signal data with time from ten channels of the system with three different hits with rotation.
(XLSX)

## Acknowledgments

The authors would like to acknowledge Xiaobo Wu, Gengsui Zhang, and Kong Ruan from Wenzhou Polytechnic for their valuable help in data collection.

## Author contributions

**Data curation:** Chao Zhang.

**Formal analysis:** Chao Zhang, Zile Fan.

**Project administration:** Yafeng Kang.

**Supervision:** Zile Fan, Yafeng Kang.

**Validation:** Chao Zhang.

**Writing – original draft:** Chao Zhang.

**Writing – review & editing:** Zile Fan, Yafeng Kang.

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
