## [Decision Letter · Decision Letter 0]

28 Jul 2025

PONE-D-25-29483Piezoelectric Row-Column Sensing System on Table Tennis Racket for Hit and Rotation MeasurementPLOS ONE

Dear Dr. Fan,

Thank you for submitting your manuscript to PLOS ONE. After careful consideration, we feel that it has merit but does not fully meet PLOS ONE’s publication criteria as it currently stands. Therefore, we invite you to submit a revised version of the manuscript that addresses the points raised during the review process.

We look forward to receiving your revised manuscript.

Kind regards,

Bonnie Gray

Academic Editor

PLOS ONE

Journal Requirements: 

2. In the online submission form, you indicated that [The data underlying the results presented in the study are available upon request from the corresponding author.].

Additional Editor Comments:

Please carefully respond to all reviewers' comments and alter the manuscript appropriately to reflect your response as outlined earlier in this letter.

Reviewers' comments:

Reviewer's Responses to Questions

**Comments to the Author**

1. Is the manuscript technically sound, and do the data support the conclusions?

Reviewer #1: Yes

Reviewer #2: Partly

Reviewer #3: Yes

2. Has the statistical analysis been performed appropriately and rigorously? 

Reviewer #1: N/A

Reviewer #2: Yes

Reviewer #3: Yes

3. Have the authors made all data underlying the findings in their manuscript fully available?

Reviewer #1: Yes

Reviewer #2: Yes

Reviewer #3: Yes

4. Is the manuscript presented in an intelligible fashion and written in standard English?

Reviewer #1: Yes

Reviewer #2: No

Reviewer #3: Yes

5. Review Comments to the Author

Reviewer #1: - Will this sensing architecture be able to distinguish between out-of-plane spin and in-plane spin and a combination of both?

- Will the hard binds affect the contact of the ball with other regions on racket where there is no bind?

- Could you explain more if the existence of binds will change the motions of ball so the measured force/spin is a distorted scenario from real scenario?

- Clearer explanation including colormap, title and legend for Fig. 6 is needed. Also please indicate the direction of force with coordinates of x and y in simulated results.

- Please include quantitative elaboration for Fig. 7 with determination of spin directions.

- Please include a comparison paragraphs between the proposed design and existing design for this motion sensing of tennis ball. Clarification on e.g., the improvements in signal transmission efficiency should be explained there.

Reviewer #2: The manuscript presents an interesting study on the development of a piezoelectric row-column sensing system integrated into a table tennis racket for hit position and rotation measurement. The authors propose a novel sensing architecture aimed at reducing the number of signal channels while maintaining high spatial resolution. The manuscript details the sensor design, experimental setup, and preliminary validation, offering potential applications in sports training and performance analysis.

While the study addresses a relevant topic and provides promising insights, several critical points require clarification and improvement before publication:

The manuscript contains multiple grammatical mistakes and awkward expressions, such as “the system is verified to monitor the rotation” and “the hitting pressure will be reduced during the transmission.” These issues affect the clarity and professionalism of the presentation.The authors are strongly encouraged to carefully revise the manuscript for language quality and scientific expression. Engaging a professional language editing service is advised to enhance readability.

Although the introduction cites several related works, the manuscript lacks a critical discussion comparing the proposed system’s performance with existing sensor-based and vision-based table tennis monitoring technologies. This limits the reader’s understanding of the actual contribution and positioning of the work within the field. The authors should add a comparative analysis—either as a dedicated paragraph or a comparison table—highlighting key aspects such as sensing resolution, system complexity, data acquisition demands, and application potential relative to state-of-the-art methods.

The concept of a row-column sensor array raises concerns about signal discrimination. Since sensors along both the row and column of an impact point are designed to produce output, it is unclear how the system ensures accurate localization of the hit without confusion from adjacent or overlapping signals, especially considering mechanical force spreading or signal cross-talk. The authors should clarify the signal processing approach used to resolve hit localization, explaining how overlapping signals are handled. Experimental evidence demonstrating the system’s ability to distinguish between closely spaced impact points would substantiate the method’s reliability.

Reviewer #3: Piezoelectric Row-Column Sensing System on Table Tennis Racket for Hit and Rotation Measurement. In this paper, a row-column sensing method is proposed to address this limitation at the sight of table tennis monitoring. I think there are still some questions that need to solve and answer before publication in PLOS ONE.

1. No comparison with prior art: e.g., 25-channel system positioning accuracy in [14] (should be quantified: ±5mm positioning error in this system vs. ±2mm in literature?); rotational measurement error in [23].

2. Unquantified power advantages: claims of “reduced hardware complexity”, but no comparative power/cost data before and after channel reduction.

6. PLOS authors have the option to publish the peer review history of their article (what does this mean? ). If published, this will include your full peer review and any attached files.

**Do you want your identity to be public for this peer review?** For information about this choice, including consent withdrawal, please see our Privacy Policy .

Reviewer #1: No

Reviewer #2: No

Reviewer #3: No

---

## [Author Response · Author response to Decision Letter 1]

1 Sep 2025

Please check the response to the reviewers file.

---

## [Decision Letter · Decision Letter 1]

18 Sep 2025

Piezoelectric row-column sensing system on table tennis rackets for hit and rotation measurement

PONE-D-25-29483R1

Dear Dr. Kang,

We’re pleased to inform you that your manuscript has been judged scientifically suitable for publication and will be formally accepted for publication once it meets all outstanding technical requirements.

Kind regards,

Bonnie Gray

Academic Editor

PLOS ONE

Additional Editor Comments (optional):

Reviewer #1:

Reviewer #2:

Reviewers' comments:

Reviewer's Responses to Questions

**Comments to the Author**

1. If the authors have adequately addressed your comments raised in a previous round of review and you feel that this manuscript is now acceptable for publication, you may indicate that here to bypass the “Comments to the Author” section, enter your conflict of interest statement in the “Confidential to Editor” section, and submit your "Accept" recommendation.

Reviewer #1: All comments have been addressed

Reviewer #2: All comments have been addressed

2. Is the manuscript technically sound, and do the data support the conclusions?

Reviewer #1: Yes

Reviewer #2: Yes

3. Has the statistical analysis been performed appropriately and rigorously? 

Reviewer #1: N/A

Reviewer #2: N/A

4. Have the authors made all data underlying the findings in their manuscript fully available?

Reviewer #1: Yes

Reviewer #2: Yes

5. Is the manuscript presented in an intelligible fashion and written in standard English?

Reviewer #1: Yes

Reviewer #2: Yes

6. Review Comments to the Author

Reviewer #1: (No Response)

Reviewer #2: Authors addressed all comments, and it can be accepted for publication ,Authors addressed all comments, and it can be accepted for publication

7. PLOS authors have the option to publish the peer review history of their article (what does this mean? ). If published, this will include your full peer review and any attached files.

**Do you want your identity to be public for this peer review?** For information about this choice, including consent withdrawal, please see our Privacy Policy .

Reviewer #1: No

Reviewer #2: No

---

## [Editor Report · Acceptance letter]

PONE-D-25-29483R1

PLOS ONE

Dear Dr. Kang,

I'm pleased to inform you that your manuscript has been deemed suitable for publication in PLOS ONE. Congratulations! Your manuscript is now being handed over to our production team.

Kind regards,

on behalf of

Dr. Bonnie Gray

Academic Editor

PLOS ONE